# The Enigma of the Adrenarche: Identifying the Early Life Mechanisms and Possible Role in Postnatal Brain Development

**DOI:** 10.3390/ijms22094296

**Published:** 2021-04-21

**Authors:** Angela L. Cumberland, Jonathan J. Hirst, Emilio Badoer, Stefan A. Wudy, Ronda F. Greaves, Margaret Zacharin, David W. Walker

**Affiliations:** 1School of Health & Biomedical Sciences, Bundoora Campus, RMIT University, Melbourne, VIC 3083, Australia; angela.cumberland@rmit.edu.au (A.L.C.); emilio.badoer@rmit.edu.au (E.B.); ronda.greaves@mcri.edu.au (R.F.G.); 2Mothers and Babies Research Centre, School of Biomedical Sciences and Pharmacy, University of Newcastle, Callaghan, NSW 2308, Australia; jon.hirst@newcastle.edu.au; 3Steroid Research & Mass Spectrometry Laboratory, Division of Pediatric Endocrinology & Diabetology, Center of Child and Adolescent Medicine, Justus Liebig University, 35392 Giessen, Germany; stefan.wudy@paediat.med.uni-giessen.de; 4Murdoch Children’s Research Institute, Parkville, VIC 3052, Australia; margaret.zacharin@rch.org.au; 5Department of Paediatrics, University of Melbourne, Parkville, VIC 3052, Australia; 6Department of Endocrinology, Royal Children’s Hospital, Parkville, VIC 3052, Australia

**Keywords:** adrenarche, DHEA/DHEAS, adrenal androgens, neurosteroids, brain development

## Abstract

Dehydroepiandrosterone (DHEA) and its sulfated metabolite (DHEAS) are dynamically regulated before birth and the onset of puberty. Yet, the origins and purpose of increasing DHEA[S] in postnatal development remain elusive. Here, we draw attention to this pre-pubertal surge from the adrenal gland—the adrenarche—and discuss whether this is the result of intra-adrenal gene expression specifically affecting the zona reticularis (ZR), if the ZR is influenced by the hypothalamic-pituitary axis, and the possible role of spino-sympathetic innervation in prompting increased ZR activity. We also discuss whether neural DHEA[S] synthesis is coordinately regulated with the developing adrenal gland. We propose that DHEA[S] is crucial in the brain maturation of humans prior to and during puberty, and suggest that the function of the adrenarche is to modulate, adapt and rewire the pre-adolescent brain for new and ever-changing social challenges. The etiology of DHEA[S] synthesis, neurodevelopment and recently described 11-keto and 11-oxygenated androgens are difficult to investigate in humans owing to: (i) ethical restrictions on mechanistic studies, (ii) the inability to predict which individuals will develop specific mental characteristics, and (iii) the difficulty of conducting retrospective studies based on perinatal complications. We discuss new opportunities for animal studies to overcome these important issues.

## 1. Introduction

The intention of this article is to discuss the possible physiological mechanisms that initiate and support the hormonal changes that characterize adrenarche in humans and some non-human primates. As such, we have not attempted a systematic review of this topic; such a review has been done expertly in recent times [1,2,3,4]. We will also suggest what could be done if a tractable, small animal model of the adrenarche could be found. Our objective is to suggest how the fundamental mechanisms that lead to the remarkable increase of steroidogenesis in the adrenal cortex in early life can be identified, and to determine what the coordinate changes in the adrenal gland and brain might be.

Adrenarche is an early endocrine event of adrenal origin in children, but how it is brought about and its significance for the subsequent development of the child remains something of a mystery. While it is generally thought that adrenarche involves a pre-pubertal increase in the secretion of the androgen dehydroepiandrosterone (DHEA) and its sulfated conjugate DHEAS from the adrenal gland in children, from about the age of six years, increasing evidence suggests that it is a much more gradual process starting from at least infancy [5,6,7], and indeed, subject to influences present in prenatal life that also affect fetal growth [8,9]. The increased secretion of adrenal androgens that essentially defines adrenarche is also thought to modulate brain development, both before and after puberty [10], suggesting a role for adrenarche in determining sex-dependent, post-pubertal behavior and phenotype.

These early-life endocrinal changes are widely considered to essentially be a human phenomenon, with only some higher-order primates undergoing a similar developmental change [11,12]—see further discussion below. While the regulation of the hypothalamo-pituitary-gonadal (HPG) axis is known in some detail, including the importance of the so-called ‘GnRH pulse generator’ within the arcuate nucleus for providing the pulsed release of LH and FSH from the anterior pituitary [13], as discussed below, the central regulation of the HPA axis that results in the selective induction of the ***zona reticularis*** (ZR) is still unclear and a matter of continuing debate. Hence, why the mechanistic understanding of adrenarche—in particular, the ***origin*** of the factors that drive the dynamic changes within a strictly defined zone of the adrenal cortex, the ZR—is still poorly defined and essentially unknown. Similarly, there has been much discussion around the impact on the brain of adrenarche and the associated increase of DHEA during human adolescence and adult life [1]. Notwithstanding the important evolutionary perspective pointed out by Campbell [1], the mechanistic aspects of adrenarche remain highly speculative, not least because of the practical and ethical limitations of conducting experiments that dissect the essential mechanisms of humans and primates. The absence of a clearly defined adrenarche in commonly used laboratory animals—rats and conventional mice synthesize little DHEA in the adrenal gland and brain [12,14,15,16]—contributes to our poor understanding of what initiates and controls the change of adrenal function before puberty in humans.

In our opinion, three important questions need to be resolved:
(1)What are the fundamental mechanisms that lead to the dramatic remodeling of the adrenal gland in early childhood? (2)Are they intrinsic to the development of the adrenal gland, or dependent on coordinated changes in the brain?(3)What are the functional consequences of adrenarche for brain development and the impacts on behavior in adolescence and early adulthood?

The following discussion is designed to review our current knowledge in relation to these questions and highlight areas that require increased knowledge to enable translation into clinical care.

## 2. Production of DHEA

Adrenarche is a process distinct in time from sexual maturation driven by the HPG axis. As a hormonal change arising from the adrenal gland, adrenarche is presumed to be controlled by the brain, but these pathways have never been adequately identified. For instance, the large increase of DHEA synthesis and secretion occurs without an increase in cortisol, so presumably, it is not solely driven by adrenocorticotropic hormone (ACTH) released from the pituitary gland; indeed, the non-ACTH regulation of the adrenal gland has been discussed at length [17], although not specifically in regard to the onset and regulation of adrenarche. There is a possibility that adrenarche results primarily from an intra-adrenal maturational event—i.e., changes intrinsic to the genetic maturation of the adrenal gland—as recently suggested [12]. But if so, what exactly are these processes, and how do they specifically involve the ZR? Do they involve changes in gene expression upstream of the genes that directly produce the proteins for steroid synthesis, or by interactions with paracrine growth factors synthesized within the adrenal cortex (e.g., IGFs, FGFs, TGFβ) that promote or inhibit enzyme activity in the ZR preferentially to the other cortical zones [18]? 

During fetal development, DHEA and DHEAS (henceforth, referred to as DHEA[S]) are synthesized in the fetal zone of the adrenal gland, the largest area of the fetal adrenal gland, which contains all of the necessary enzymes required for C_19_ steroid production [19]. Postnatally, the fetal zone undergoes apoptosis and inversion before developing into the mature ZR of the adult adrenal gland [20]. The key steps in the steroidogenic pathway for the synthesis of DHEA are shown in Figure 1. 17-OH-pregnenolone is the essential immediate precursor for conversion to DHEA, and the co-presence of the enzymes CYP17A1 and Cyt-B5 within the corticotroph cells in the ZR is required for the lyase reaction to occur for the formation of DHEA. The abundance of 17-OH-pregnenolone is also crucial, and this depends not only on the hydroxylase activity of CYP17A1, but also on 3β-HSD1/2 expression and activity, which uses 17-OH pregnenolone as the substrate for the glucocorticoid pathway; i.e., for cortisol or corticosterone production. The factors that regulate the differential expression and activity of these key enzymes are unknown, as indeed are the factors that induce the hypertrophy of the ZR and not the other cortical zones before the onset of puberty.

Recent MS-based steroidomics have revealed that 11-oxygenated androgens are highly expressed in humans and many other species [21], and 11-ketotestosterone is the dominant bioactive androgen in children [22]. These findings will result in a significant revision in thinking about the mechanisms leading to adrenarche and may allow the extension of experimental work to non-primate species, for example, the high levels of 11-oxyandrogens present in mature pigs and guinea pigs [21]. The guinea pig is particularly interesting as, like rodents, little DHEA or DHEAS is produced by the adrenal gland, and there is no maturation or puberty-related changes in these steroids. 11-ketotestosterone is a potent androgen, and while it is now of interest to the physiology of human adrenarche, the action of this androgen in the guinea pig remains unclear and levels in immature guinea pigs are yet to be determined. Importantly, in guinea pigs, the levels of 11-keto and 11beta-hydroxy testosterone and androstenedione produced by the ZR are markedly upregulated by ACTH, suggesting that plasma levels are under the control of the HPA axis [23]. It remains unclear whether the guinea pig undergoes adrenarche-like changes in these androgens during postnatal maturation, and if the spiny mouse—which does have a pre-pubertal surge of peripheral DHEA [24]—also synthesizes and releases the 11-oxygenated androgens now known to be associated with adrenarche in children. Also unknown for any species is the full spectrum of activity of these particular androgens in the brain, a line of inquiry that is important to firmly establish whether adrenarche is a developmental event that determines the trajectory of brain development through and beyond puberty.

Conventionally, the postnatal but pre-pubertal increase in the adrenal production of DHEA and DHEAS has been viewed as a phenomenon restricted to humans and some non-human primates (chimpanzee, bonobos, gorillas) [25,26], but a small precocial rodent proves an exception [24,27]. Also, the recent analysis of the steroid metabolome using LC/MS referred to above has revealed the presence of perhaps physiologically relevant peripheral concentrations of DHEA in rabbits, dogs and cows (~1 nmol/L), and of DHEAS in pigs (~0.04 umol/L), together with a number of 11-oxyandrogens [21] (see Figure 1). However, the ontogenic changes of DHEA/S and the 11-oxyandrogens are yet to be defined, and the possible impact of the latter on brain development is currently unknown. This is important because it may substantially refine our understanding of the role of steroids in brain development before and after puberty.

DHEA is readily sulfated in both adrenal and brain tissue, and the release of DHEAS from the adrenal gland greatly exceeds that of DHEA [28]. The half-life of DHEAS in the circulation is far longer (>5 h) than for DHEA (15–30 min), and because DHEAS is readily back converted to DHEA, a distinct function of DHEAS may be to act as a reserve pool for DHEA, thereby extending the physiological availability of DHEA. Although manifold steroid transporters have been identified at the blood-brain barrier and the choroid plexus [29], the sulfation of DHEA retards its movement from the circulation to peripheral tissues, and DHEAS does not readily cross the blood-brain barrier [30]. The ‘de novo’ synthesis of DHEA in the brain may, therefore, be particularly important, but how this changes postnatally and before puberty is unclear. 

## 3. Why Is This Important?

In addition to the adrenal cortex, DHEA synthesis occurs in neural tissue in some species [20,27]. However, the relationship between the adrenal and brain synthesis of DHEA is relatively unexamined, and raises the interesting question—does the maturational drive that increases adrenal DHEA synthesis also apply to the central nervous system (CNS)? The potential role of DHEA[S] in the evolution of brain development and sexuality during childhood and pre-adolescent development, and in determining the cognitive and behavioral orientation of boys and girls, are ideas of great interest to socio-biologists and pediatric endocrinologists. But this remains a highly speculative debate as a result of lack of evidence due to the obvious difficulty of performing mechanistic studies in humans. DHEA and DHEAS appear to have separate modulatory activities in regulating neurite growth and shaping network projections in the brain; for example, DHEA potentiates neurogenesis, neuronal survival, axonal growth and synaptogenesis, whereas DHEAS promotes dendritic growth and branching [31]. Gonadal hormones clearly play a role in determining the function of the hippocampus directly and indirectly via cholinergic projections from the septohippocampal gyrus (reviewed in [32]). DHEA treatment itself induces an increase of synaptic spine density in the hippocampus of the rat [33], although it is important to note this effect of exogenous DHEA occurs in a species that synthesizes little of this androgen.

It has been hypothesized that a primary role of DHEA, whether of adrenal or central origin, is to function as a neuroactive steroid. A number of studies support roles for DHEA[S] in various aspects of neuro activity, raising the possibility that a key function of the adrenarche is to begin to modify the neural, behavioral, and psychosocial development that is so characteristic of puberty and adolescence, although exact mechanisms currently remain unclear. That is, the presence of high levels of DHEA[S] before puberty may allow for regional and/or sex-specific brain development before the commitment of the brain to the events that occur at puberty.

DHEA and DHEAS interact with many neurotransmitter receptors, including the sigma (σ), glutamate and GABA_A_ receptors [34,35]. DHEA may also protect against the effects of glucocorticoid neurotoxicity [36]. In vitro, the toxic effect of corticosterone in the dentate gyrus of the hippocampus of male rats is suppressed by even low concentrations of DHEA [37]. Furthermore, this effect appeared to be specific for DHEA, because neither the steroid precursor (pregnenolone) nor a closely related androgen (3β,17β-androstenediol) had anti-glucocorticoid effects. In turn, such data support the commonly held idea (but also not well supported by mechanistic evidence) that DHEA[S] has anti-aging effects (the ‘hormone of youth’), because stress produces prolonged exposure to high levels of circulating glucocorticoids and this causes atrophy (neurodegeneration) of certain brain pathways, especially of memory-related hippocampal neurons [38]. The exact mechanism(s) for the anti-glucocorticoid action of DHEA remains unclear. Is it possible that a key function of the adrenarche is to modify the neural, behavioral, and psychosocial development that is characteristic of puberty and adolescence?

There is good evidence that DHEA also reduces other types of excitotoxicity. In vitro, DHEA protects immortalized mouse hippocampal HT-22 cells against glutamate and β-amyloid protein toxicity in a dose-dependent manner [39], and both DHEA and DHEAS protect cultured fetal rat hippocampal neurons against NMDA, AMPA and kainic acid toxicities [40]. This neuroprotection does not appear to involve direct interaction with glutamate receptors, and appears to be facilitated via alternative pathways, such as the σ_1_ receptor [41] or by protecting mitochondria against the effects of high intracellular Ca^2+^ [42], or again, by modulation of the calcium/nitric oxide (NO) signaling pathway [40]. Following the comprehensive analysis given to pregnanolone-derived neurosteroids in the adult brain [43], it is possible that in the postnatal brain, DHEA is important to protect nerve fibers and oligodendrocytes against glucocorticoid-mediated neurotoxicity, particularly in white matter tracts and pathways associated with sensory inflow and motor outflow to the cerebellum and spinal cord. Local DHEA synthesis would also lead to the rapid production of DHEAS, which, in the developing neocortex, acts to promote dendritic growth and branching, whereas DHEA promotes axonal growth and synaptogenesis [31]. Thus, DHEA and DHEAS could have separate modulatory activities in regulating neurite growth and shaping network projections [34]. Because DHEAS does not readily cross the blood-brain barrier, the ‘de novo’ synthesis of DHEA in the brain in the presence of the DHEA sulfotransferase may be particularly important. However, attention has been drawn to the presence of uptake and efflux transporters for steroids in the brain, including at the blood-brain barrier, choroid plexus, and the possibility of interchange between glia (astrocytes, in particular) and neurons [29]. The importance of these transporters (comprising members of the ATP-binding cassette [ABC], solute carrier-type [SLC] and organic anion transporting polypeptide [OATP] families) for the developing brain is poorly understood, and their role in determining the entry and intra-cerebral regulation of neurosteroids, in general, and DHEA, in particular, need to be investigated further. Each of these transporters has wide and overlapping substrate specificities, and individually none may be critical in determining steroid concentrations in the brain’s extracellular or cerebrospinal fluids. Whether changes in the expression and activity of these transporters occur concurrently with adrenarche is yet to be investigated.

## 4. Relationship of DHEA[S] with Psychiatric Disorders and the Potential Developmental Origins of Abnormal Adrenarche

DHEA[S] has long been considered in the treatment of neuropsychiatric disorders, with many studies reporting abnormal serum DHEA concentrations in patients with major neurodevelopmental and neurodegenerative pathologies, including schizophrenia, bipolar affective disorder, depression and Alzheimer’s disease [40,41]. In relation to Alzheimer’s disease, a decrease [44], increase [45] or no change in DHEA[S] [44,46] has been reported in association with disease symptomology. Schizophrenia and schizoaffective disorders have been associated with reductions in circulating DHEA[S] compared to levels found in individuals without schizophrenia [47,48,49]. Conversely, increased levels of DHEA have been reported in individuals diagnosed with post-traumatic stress disorder [50,51,52,53]. The data around DHEA[S] in depression are conflicting, with some reports finding reductions in DHEA with remission of depressive symptoms [54], while others suggest that individuals with greater circulating DHEA pre- and post-antidepressant treatment are more likely to see improvements in negative symptoms [55]. Despite these discrepancies in DHEA[S] associated with neurological disorders, a strong link remains between circulating levels of DHEA[S] and these mental health conditions. Yet, the mechanistic evidence for DHEA or DHEAS having a defined role in any of these domains in human development remains speculative, not least because of the ethical and practical limitations noted above. 

Abnormal adrenarche, primarily premature adrenarche (PA), as measured by increased serum DHEA (>1 µmol/L) before the ages of eight and nine in girls and boys, respectively [56], has been linked to the development of psychiatric disorders such as depression, anxiety and externalizing or aggressive disorders [57]. There is also a sex difference reported in the presentation of psychiatric disorders in children with PA [56]. Marakaki et al. found increased reporting of anxiety and depression scores in girls with PA in the absence of salivary cortisol or hypothalamic-pituitary-adrenal (HPA) dysregulation, compared to girls with on-time adrenarche, while there was no reported difference between boys with and without PA [58]. Sontag-Pallida et al. also reported that serum cortisol levels and executive function in PA were associated with the presentation of differing mood and behavioral disorders in girls [59]. These authors reported that girls with PA and low executive function were more likely to present with externalizing and anxious symptoms, compared to girls with PA and higher executive function or girls with on-time adrenarche. Serum cortisol levels in PA were associated with different symptoms, with low levels associated with depressive symptoms and high serum cortisol with higher externalizing symptoms [59]. PA is also associated with an increased risk of developing polycystic ovarian syndrome [60,61], of which depression and anxiety disorders are common co-morbidities [62,63,64]. There are also reports of an increased risk of cardio-metabolic disorders associated with PA [65]. However, this may be due to children who present with PA being more likely to be overweight or obese [66]. These data emphasize, with or without concurrent altered cortisol release, an early onset of adrenarche, as measured by the premature surge of DHEA[S] from the adrenal gland, is associated with altered long-term reproductive and neurodevelopmental outcomes, and warrants attention in understanding how this event may result in the onset of mental health disorders.

The development of diseases in later life, including psychiatric disorders, can also be linked to the perinatal events now grouped under the developmental origins of health and disease (DOHaD) hypothesis [67,68]. Indeed, PA has been suggested to have an early developmental origin. Being born with low birth weight is one of the leading factors linked to the DOHaD hypothesis, with intrauterine growth restriction (IUGR) being the second-leading cause of low birth weight in infants after prematurity [69,70]. IUGR, the pathological restriction of fetal growth, is a known confounder of fetal brain development and causes long-term dysregulation of the HPA axis in both humans and animal models [71,72,73,74]. DHEA is reduced in the IUGR neonate [75,76], suggesting an impairment of adrenal production following compromised in utero growth. Third-trimester fetuses with IUGR have increased adrenal gland volume compared to appropriately grown fetuses, but have reduced fetal zone volume [77]. The authors postulate that this is due to altered activation of the HPA axis and increased cortisol output from the fetal adrenal gland, or due to fetal blood flow redistribution in IUGR to maintain adrenal perfusion and hence function [78,79]. The decrease in fetal zone volume aligns with reductions in DHEA output from the impacted adrenal glands [80]; however, whether this is an adaptive mechanism in which the fetal adrenal sacrifices DHEA synthesis for cortisol output to support lung and heart development, with a later impact on postnatal adrenal function, is yet to be addressed. Overall, whether premature adrenarche and/or the associated disorders are a cause or consequence of impaired adrenal secretion, or CNS synthesis and utilization of DHEA[S], is difficult to discern due to the limitations in modeling the pre-pubertal DHEA[S] surge, as mentioned above.

## 5. A New Approach to an Old Problem

The significance of the pre-pubertal surge of DHEA synthesis in some species, primarily humans and some other primates, and how it comes about, is yet to be fully explained. Clearly, known hormones of pituitary origin (e.g., kisspeptin, GH, FSH, LH), in addition to ACTH, could be tested, and the presence of a cortical activating synthesis hormone [CASH] has long been suspected but never identified. The ready availability of adrenal glands from an animal that does secrete DHEA might provide the opportunity to provide definitive proof of the presence of a CASH hormone that specifically activates the DHEA pathway in postnatal ZR. Figure 2 outlines the known and putative drives that may impact the adrenal gland, including the accepted idea of pituitary ACTH (though possibly at low constant levels), splanchnic innervation of both the adrenal medulla and cortex, and intra-adrenal paracrine factors. The splanchnic nerve densely innervates the adrenal medulla, but whether some fibers extend into the adrenal cortex and influence the activity of the ZR, in particular, remains an important topic of discussion. Surprisingly, it is not known if pre-pubertal adrenal glands isolated from a DHEA-secreting species respond to muscarinic and nicotinic receptor agonists. The question being raised here is whether the autonomic nervous system, perhaps in concert with the hypothalamic-pituitary axis, is involved in inducing the childhood activation of adrenal ZR. The possibility of innervation of the adrenal cortex via the splanchnic nerve [81], and the anatomical evidence for the presence of adrenomedullary tissue throughout the adrenal cortex [82], suggest that descending sympathetic activation from CNS networks might be involved in the onset of adrenarche. Also, there is the possibility that the adrenal gland is itself a source of ACTH [83] and other neuropeptides (reviewed in [84]), and that splanchnic nerve stimulation leads to increased cortical sensitivity to ACTH (at least, for cortisol) [83], leading to the reasonable speculation that adrenocortical secretion of androgenic steroids can occur via a non-pituitary pathway (see [17] for review). 

There is the important question of what impact the adrenarche might have on brain development and function, both for normal development and following an abnormal adrenarche event. Does increased androgen production before puberty affect neuroplasticity, giving adaptability (plasticity) in behavior, learning and memory, which might differ between the sexes? The possibility that DHEA protects the adult brain against the effects of chronic stress involving glucocorticoids has been widely discussed [37,86]. While important species differences exist in the distribution of the glucocorticoid receptor (GR), particularly in the hippocampus [87,88], for many species, glucocorticoid signaling in the brain is important throughout life, and DHEA may be an important endogenous regulator of this pathway. The extent to which peripheral (i.e., adrenal) or centrally-derived DHEA concentrations can affect the expression and activity of the GR in the brain also requires further investigation. The recent observation that there is a concurrent (coordinated?) shift of CYP17A1 and GR expression during development, from a ubiquitous distribution in the fetal brain to a largely restricted expression in the white matter of the adult brain [24], suggests a functional coupling between glucocorticoid signaling and the localized synthesis of DHEA[S]. There is also a shift of DHEA synthesis from neurons to glia in the transition from fetal to postnatal life [24].

## 6. Conclusions: What Needs to Be Done Now?

We suggest that four goals need to be met to help resolve the enigma of the adrenarche, and to understand the impact it might have on postnatal brain development. Firstly, inquiry into the early development of animals that actually have adrenal synthesis of DHEA may provide a comparative animal model of adrenarche. We have recently discovered that a precocial rodent—the spiny mouse *(Acomys cahirinus)*—has a pre-pubertal DHEA surge underpinned by coordinated changes in the structure and functional activity of the ZR of the adrenal gland [24,27], and in this precocial species DHEA is also synthesized ‘de novo’ in the brain [24]. This gives the opportunity for well-designed animal studies to provide new insights into these important questions. For instance, if the hypothalamus-pituitary axis is indeed involved in triggering the pre-pubertal surge of DHEA from the adrenal gland, then a pituitary factor should be present in the blood at this time. However, further examination in other animals is warranted, particularly in view of the recent description of the additional 11-oxyandrogens of adrenal origin [21].

Secondly, gene expression studies could then identify changes of expression of the entire range of genes involved in cholesterol utilization by the adrenal gland, and perhaps identify the key transcriptional co-factors that regulate the growth and activity of the ZR.

Thirdly, a comprehensive steroid metabolomics approach is required to track changes that occur during postnatal development in such animals, comparable to detailed analyses recently completed for humans [22,89,90,91] and some other species [21].

Finally, how can we settle the question of whether a circulating factor is responsible for developmental changes in the adrenal gland and the postnatal brain? One approach might be to isolate the pre-adrenarchal, undifferentiated adrenal gland from a species that has ‘adrenarche-like’ changes in early-life, maintain it in culture ex vivo and then expose it to plasma (and/or plasma extracts) obtained from progressively more mature individuals of the same species, to determine if a blood-borne factor is able to induce subsequent differentiation and maturation of the ZR. In a species such as the spiny mouse, where the adrenarche-like changes occur over a short period (i.e., 10–20 days), maintaining the isolated adrenal gland without external stimulation would also allow the identification of maturational changes that are truly intrinsic to the adrenal gland, as suggested elsewhere [12,18]. Additionally, maintenance of postnatally adrenalectomized animals would help to determine if adrenal factors are also important in shaping brain development pre- and post-puberty.

To conclude, resolving the enigma of the adrenarche has proved difficult because of the absence of an experimentally tractable animal in which mechanistic studies can be performed. Closer examination of the postnatal development of some species that do synthesize androgens in both the adrenal gland and brain may be fruitful.

## Figures and Tables

**Figure 1 ijms-22-04296-f001:**
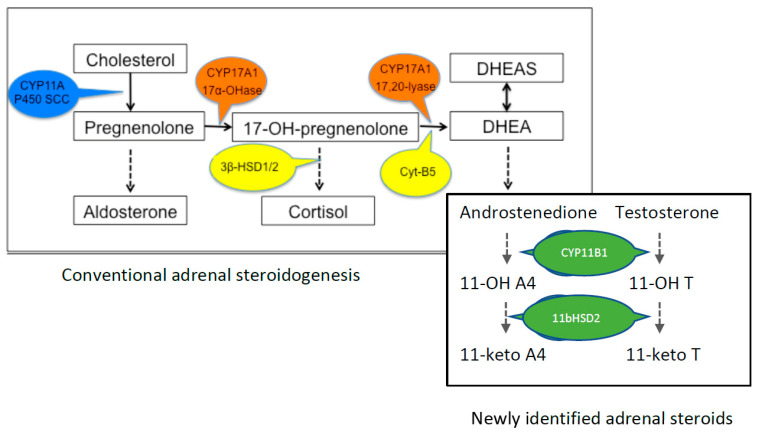
Key parts of the steroid pathway in relation to DHEA synthesis. For convenience, the downstream precursor components of aldosterone, cortisol and sex steroids’ synthesis are not shown. The enzymes critical for determining the availability of 17-OH-pregnenolone for DHEA synthesis are shown in the colored bubbles. The enzyme proteins are: CYP, cytochrome P450; HSD, hydroxysteroid dehydrogenase; Cyt-B5, cytochrome B5. Steroids: A4, androstenedione; T, testosterone.

**Figure 2 ijms-22-04296-f002:**
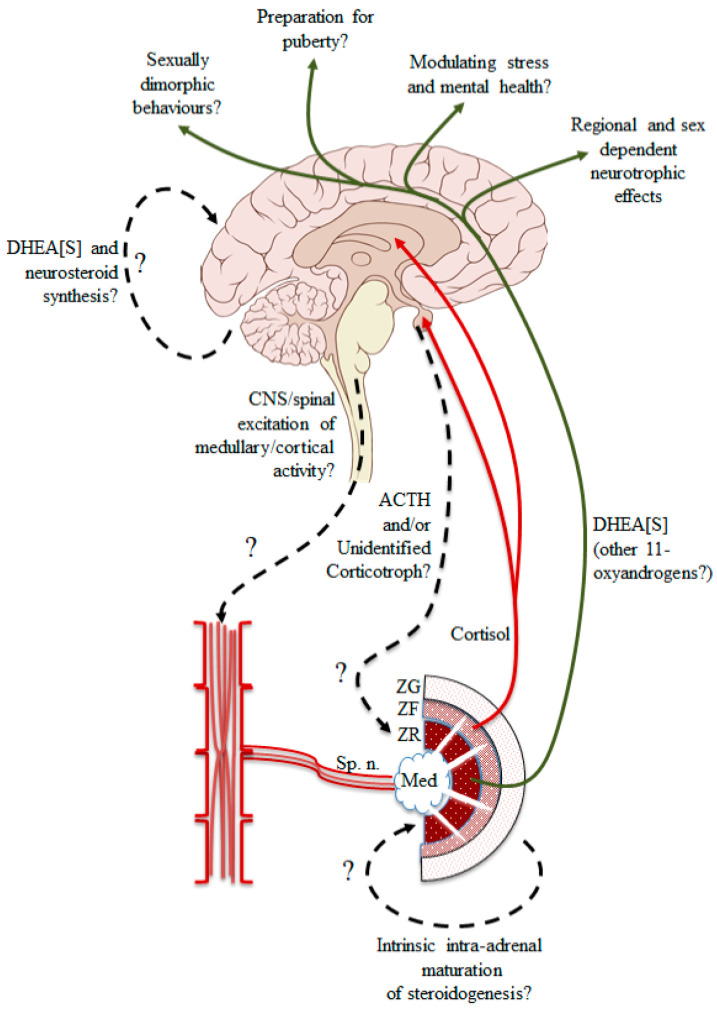
Summary of the putative descending pathways impacting the adrenal gland at adrenarche (descending spinal-sympathetic drive, pituitary derived ACTH and/or an unidentified corticotropic substance), together with the intra-adrenal maturation of paracrine influences on the ZR. The established effects of DHEA[S] on the brain are shown, with the recently described 11-oxyandrogens as possible co-modulators of brain activity and development. Coordinated changes of adrenal and CNS DHEA[S] synthesis are also suggested. Med = medulla. Brain image adapted from [85] under the *Creative Commons Attribution 2.5 License 2006*.

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
