# Peer review of "The Enigma of the Adrenarche: Identifying the Early Life Mechanisms and Possible Role in Postnatal Brain Development"

_ijms, 2021, doi:10.3390/ijms22094296_

Round 1
Reviewer 1 Report
This review tried to summarize the recent finding on the possible role in postnatal brain development exerted by adrenarche. Authors concentrated the action of DHEA and DHEAS, which is mainly secreted from adrenal gland, on the brain development during puberty. They also pointed out that the difficulty in study this field in human being is due to ethical restriction, unpredictable mental characteristics, and the difficulty of conducting retrospective studies from preinatal complication.
Comments
- What is the difference between HPA and HPG on puberty? It is necessary to clarify these difference. Also, it should be appeared in “Introduction”.
- In the context, DHEA and DHEAS are responsible for the puberty in brain development, mainly on the psychological and metal characteristics. Authors might add some examples of psychological and mental characteristics.
- A summarized figure could better for reader to read and understand.
Reviewer 2 Report
Adrenarche and its role in postnatal brain development is a timely and important topic for a review. The manuscript presents a lot of interesting ideas and questions to be answered, but the organization of the paper is confusing and leaves the reader with more questions than answers. This manuscript is organized like a proposal to use the spiny mouse as an animal model to study adrenarche, rather than a "more typical" review of what is known about adrenarche and postnatal brain development. In many sections (production of DHEA; why is this important?) this paper does not provide a current review of the literature and it lacks citations throughout the paper. The lack of scientific references makes it unclear when the authors are theorizing about how things might work and when they are stating what is known from the scientific literature. I have outlined my issues and clarified my concerns in the comments below.
MAJOR COMMENTS
- The organization of the paper is confusing. The paper is organized into six sections: (1) Introduction; (2) Production of DHEA; (3) Why is this important?; (4) Developmental origins of abnormal adrenarche?; (5) A new approach to an old problem; and (6) Conclusions: what needs to be done now?. The paper reads more like a proposal to use the spiny mouse as an animal model, instead of a more typical review paper about what is known about adrenarche and postnatal brain development (as suggested by the title of the paper).
- The manuscript does not provide a current review of the literature. I expected that a review of how adrenarche could affect postnatal brain development would provide a review of the actions of DHEA and DHEAS in the brain. The manuscript includes older studies of DHEA(S) actions on neurodevelopment, anti-glucocorticoid and neuroprotection (lines 110-162), but not a current review of the literature. Surprisingly, there is also no mention of the conversion of DHEA into more potent androgens and estrogens affecting development through androgen or estrogen receptors (in the brain or in the body). This should be included in a review of possible actions that DHEA(S) could be having in the brain.
- The manuscript lacks citations throughout the text. Statements are made throughout the manuscript without any references (see minor comments below). More references would help this paper be more of a review of the scientific literature than an opinion paper. It is unclear in some cases when the authors are theorizing about how things work and when they are summarizing what is known.
- The authors should review the literature on the interactions between the adrenal medulla and the adrenal cortex, i.e. intra-adrenal regulation. Intra-adrenal is mentioned on line 72, but the authors do not go into any details. As the authors note on line 225, the adrenal medulla is innervated (and thus is another way that the central nervous system can communicate with the adrenal gland).
- The review of species differences should include nonhuman primates. The manuscript jumps back and forth between the human literature and the rodent literature, and skips the nonhuman primate literature. This is important if the authors are setting up the context for a new comparative animal model – the spiny mouse. (a) This review would be stronger if it included what is known in nonhuman primates. This could help bridge the gap between what is hypothesized and known about altricial humans (and a long adolescence) and precocial rodents. (b) Given the amount of species differences in secretion of DHEA(S), some statements need clarification of what species the statements apply to (see minor comments below).
- Be more careful with relationship between DHEA(S) and psychiatric disorders. The authors conclude that the premature adrenarche (PA) could lead to “detrimental long-term neurodevelopmental outcomes” (lines 185-189). This sentence should be changed. It is unclear from the data what the direction of the relationship is – is the PA causing the psychiatric condition, or is the psychiatric condition causing the PA, or is some other factor causing both PA and a mental illness? Abnormal serum concentrations of DHEA or DHEAS in adult patients with psychiatric disorders are typically LOWER, not higher, than patients without disorders. In the next paragraph (lines 205-210), the authors rightly conclude that PA and/or the associated disorder could be a “cause or a consequence.”
MINOR COMMENTS
- Introduction, lines 38-40. Does adrenarche prime the HPG axis? This needs a reference.
- Introduction, third question (lines 59-60). “What are the functional consequences of adrenarche for brain development and the emergence of gender-related behaviours in adolescence and early adulthood?” is not addressed later in the paper and should be deleted.
- Production of DHEA, lines 65-77. It would be helpful if the authors explained the events that lead to DHEA secretion before speculating on how ACTH is not the only driver of adrenarche. Need more references here.
- Lines 81-82. Need reference for how fetal zone develops into zona reticularis.
- Lines 99-100. Which species convert DHEA to DHEAS using steroid sulfatase in adrenal and brain tissue? Need reference.
- Lines 100-103. Need to mention steroid sulfatase. Need reference.
- Lines 110-111. Clarify which species synthesize DHEA in neural tissue. Need reference.
- Line 126. Clarify androgen “from adrenal cortex”, since rats synthesize DHEA in neural tissue. Need reference.
- Line 115. What is the role of DHEA(S) in determining sexual orientation? Include reference.
- Line 153. What are endocrine-induced stresses? Include reference.
- Lines 190-192. This first sentence in the paragraph is confusing and needs a reference. Is a perinatal event a developmental origin of health and disease? Not all infants born with low birth weight have intrauterine growth restriction (IUGR).
- Lines 212-213. Clarify which species have pre-pubertal surge. Need reference.
- Lines 225-226. Need reference.
Round 2
Reviewer 1 Report
OK!
Reviewer 2 Report
This review, commentary and discussion of the mechanisms of adrenarche and its potential role in postnatal brain development is a timely and important contribution to the literature. The updated introductory paragraph nicely clarifies the goals of the paper. The authors have very thoughtfully and fully addressed all of my and the other reviewer’s comments on the original submission of the manuscript. The revised sections of the paper with additional references are clear and helpful updates to the paper. The addition of Figure 2 is great. I think this manuscript is a helpful contribution to our understanding of the mechanisms of adrenarche, as well as what we do not know and future directions of research. With more researchers being interested in the mind and behavioral changes that occur during the “adrenarche period” of human development, this paper will be a very helpful reference for researchers to gain a better appreciation of the adrenal changes and potential mechanisms for how DHEA(S) is acting in the brain.
Recommend: Accept in present form